# Indirect associations of pain resilience and kinesiophobia with the relationship between physical activity and chronic pain

Nils Georg Niederstrasser[ID][1]*, Nina Attridge[1], P. Maxwell Slepian[2,3,4]

**1** Department of Psychology, Sport, and Health Sciences, University of Portsmouth, Portsmouth, United Kingdom, **2** University of Toronto Centre for the Study of Pain, University of Toronto, Toronto, Ontario, Canada, **3** Department of Anesthesia and Pain Management, Toronto General Hospital, Toronto, Ontario, Canada, **4** Department of Anesthesiology and Pain Medicine, University of Toronto, Toronto, Ontario, Canada

* nils.niederstrasser@port.ac.uk

## Abstract

### Background

Pain is associated with a decrease in physical activity for most individuals. Nevertheless, some individuals manage to maintain physical activity levels despite pain. While the exact psychological mechanisms behind this are unknown, it may possibly be due to low kinesiophobia and high pain resilience levels. This study aimed to examine the direct and indirect associations of pain resilience and kinesiophobia with the relationship between pain and physical activity.

### Methods

In this cross-sectional study data were collected from 172 participants suffering from chronic pain. Three path models were fitted to assess the indirect associations between pain resilience and kinesiophobia in the relationship between physical activity and musculoskeletal pain individually and simultaneously. Additionally, a linear regression model was fitted to examine the impact of psychological predictors of physical activity while accounting for musculoskeletal pain.

### Results

Significant proportions of the association between musculoskeletal pain on physical activity occurred through both pain resilience and kinesiophobia. Nevertheless, when examined simultaneously, only the indirect associations via pain resilience remained significant. Similarly, when predicting physical activity levels, only high levels of pain resilience and male gender were associated with increased physical activity levels, whereas kinesiophobia was not.

**Data availability statement:** The data, companion document, and analysis script are available from the University of Portsmouth Research Portal, which functions as a data repository: https://researchportal.port.ac.uk/en/projects/the-role-of-pain-resilience-in-the-association-between-physical-a.

**Funding:** The author(s) received no specific funding for this work.

**Competing interests:** The authors have declared that no competing interests exist.

## Conclusions

This highlights the central role pain resilience plays in retaining physical activity levels when faced with chronic pain. It also implies that pain resilience predicts physical activity levels beyond pain intensity, kinesiophobia, pain duration, and pain spread. It is, therefore, imperative to examine avenues of increasing pain resilience among individuals suffering from chronic pain, not only to improve their pain, but also their overall health and well-being. This possibly bears implications for clinical practice and may inform treatment approaches, whereby pain resilience may be boosted to increase physical activity levels. Nevertheless, given the cross-sectional design, longitudinal and experimental studies are needed to confirm the causal pathways.

## Introduction

Physical activity is widely recognized for its role in managing chronic pain. Regular physical activity can be protective against pain [1,2], whereas inactivity is associated with an exacerbation of pain complaints [3]. Alongside this, physical activity is associated with a host of additional benefits, including improved physical function and enhanced mood [4–6]. Despite these advantages, individuals with chronic pain often struggle to remain physically active due to their pain as well as the fear and avoidance associated with it [7].

Whereas the relationship between pain and physical activity is well-established, little is known about the underlying psychological mechanisms that influence this association. Historically, researchers have devoted considerable attention to understanding how negative psychological factors heighten susceptibility to pain-related distress, disability, and poor mental health [8], focusing in this context mainly on concepts such as kinesiophobia (fear of movement-related pain [9]), e.g., [10,11]. This fear stems from the belief that movements will lead to worse pain or cause harm, leading to activity avoidance and safety behaviours to avoid pain [12]. Research has shown that high levels of kinesiophobia are associated with lower levels of physical activity and worse physical function, especially among individuals suffering from conditions such as osteoarthritis and fibromyalgia [11,13]. Furthermore, kinesiophobia is associated with greater pain intensity, as confirmed by numerous studies among chronic pain patients [14,15] and so acts as a psychological barrier that helps explain how and why pain often leads to reduced physical activity.

Although the literature has historically emphasised maladaptive psychological processes in pain, incorporating the study of positive factors offers a complementary perspective that may advance theoretical understanding and inform more holistic interventions. Recently, positive psychological factors have garnered increasing attention, as newer clinical approaches place greater emphasis on fostering positive mental and physical well-being even in the presence of persistent pain, as, for example, in the context of Acceptance and Commitment Therapy [16]. Resilience, in particular, is such a concept that has drawn growing interest in pain research,

and pain resilience is the ability to maintain positive physical and emotional functioning despite pain [17]. For example, high levels of pain resilience have been shown to be prospectively associated with better mental and physical health among low back pain sufferers [18]. Further, individuals with high pain resilience have been suggested to be better able to sustain engagement in daily activities, experience less psychological distress, and report a lower intensity of pain compared to those with low resilience [19,20]. It is likely that resilient individuals are less likely to interrupt physical activity due to pain and therefore should be less affected by the negative effects of sedentariness. It remains unclear whether this also influences pain-physical activity relationship. Importantly, pain resilience is considered a modifiable trait, suggesting that interventions aimed at enhancing resilience could play a critical role in promoting physical activity in individuals with chronic pain [16]. By enhancing resilience, individuals might be able to better manage their pain and maintain physical activity, reaping the rewards associated with physical activity engagement. The current study therefore aimed to critically examine whether pain is indirectly associated with physical activity through pain resilience among a diverse group living with chronic pain.

## Methods

### Study design and settings

This was a cross-sectional online study. Participants were recruited via the online research participation platform Prolific (www.prolific.com) between 05/02/2024 and 13/02/2024. Once participants had indicated their willingness to take part in the study on the Prolific platform, they gave informed consent and, in random order, completed the FRAIL Scale, the Pain Resilience Scale (PRS), the Recent Physical Activity Questionnaire (RPAQ), the Tampa Scale of Kinesiophobia (TSK), rated their average and worst pain in the last seven days, and provided details about their pain (i.e., duration, possible diagnosis, and location). The survey was hosted on the online research platform Gorilla (www.gorilla.sc). Participants provided demographic information at the end of the survey (i.e., age and gender).

### Participants

All study procedures were approved by the University of Portsmouth Ethics Committee (SHFEC 2023−090). Written informed consent was obtained from all participants prior to taking part in the study.

Inclusion criteria were: over eighteen years old and suffering from chronic pain, i.e., pain lasting or recurring for more than 3 months, whereas participants were excluded from participating if they suffered from uncorrected vision impairment. A total of 211 participants started the Gorilla survey (www.gorilla.sc). Thirteen participants were excluded due to failing the attention check, 15 participants were excluded as they reported suffering from pain for fewer than three months and therefore not qualifying as chronic pain, ten participants were excluded due to missing data, and one participant was excluded due to reporting impossible data, i.e., suffering from pain for 100 years (despite being only 52 years old). The final study sample consisted of 172 participants (86 females (50.0%), 83 males (48.3%), 2 non-binary/third gender (1.2%), 1 not reported (0.6%)). Participants' mean age was $M_{age} = 41.92$, $SD_{age} = 13.21$; median = 39.00; range: 21–75 years.

### Measures

**Pain assessment.** Average and worst pain intensity in the past seven days were measured using visual analogue scale (VAS), which was a ten-centimetre line on which participants marked what best fits their pain intensity [21,22]. The line was anchored by "no pain" on the left and "worst pain" on the right. Next, participants were asked to report a diagnosis for their pain condition if known, and to report the duration of their pain. Finally, participants indicated where on their body they experience pain by selecting from the following: head, neck, face, shoulder, chest, abdomen, upper leg, lower leg, foot, upper back, mid back, lower back, and buttocks. Participants could select as many locations as applied.

**Tampa Scale for Kinesiophobia.** The Tampa Scale for Kinesiophobia (TSK) quantifies fear of movement and re-injury [23,24]. Kinesiophobia is an irrational and/or excessive fear of movement or physical activity that leads to avoidance of feared movements. The scale consists of 17 statements, such as "I am afraid that I might injure myself accidentally" or "My pain would probably be relieved if I were to exercise". Respondents indicate their agreement to the statements on a 4-point Likert scale, ranging from "strongly disagree" to "strongly agree". Scores range from 17 to 68 and higher scores indicate greater levels of kinesiophobia.

**Pain Resilience Scale.** The Pain Resilience Scale [25] assesses resilience in the face of pain. In this context, resilience is defined as upholding positive emotional and physical functioning when facing physical or psychological challenges, such as pain. Participants rate their agreement to 14 statements on a 4-point Likert scale, ranging from "strongly disagree" to "strongly agree", relating to their behaviour and thoughts when faced with pain. Examples include "When faced with pain I get back out there" and "When faced with pain I avoid negative thoughts". Higher scores indicate greater levels of pain resilience.

**Frailty assessment.** The FRAIL scale comprises five self-reported items: Fatigue, Resistance, Ambulation, Illness, and Loss of weight [26]. Fatigue was assessed by asking respondent to indicate how much of the time in the last four weeks they felt tired on a 5-point scale, ranging from "none of the time" to "all of the time". Participants score one point if they respond "all of the time" or "most of the time". Resistance refers to the difficulty walking up 10 steps without the use of aids and without resting. Participants respond either yes or no whereby "yes" scores an additional point towards their total frailty score. Similarly, ambulation was assessed by asking participants whether they had difficulty walking several hundred yards without aids. To assess illness, participants indicated which of the following illnesses apply to them: hypertension, diabetes, cancer, chronic lung disease, heart attack, congestive heart failure, angina, asthma, arthritis, stroke, and kidney disease. If five or more illnesses are reported a score of one is added to the total score. Loss of weight was quantified by checking whether there was decline in participants' weights of 5% or more in the past twelve months. A decline of 5% or greater means another score of one is added to the total. Total scores thus range from 0 (best) to five (worst), whereby a score between three and five represents frailty, one and two pre-frailty, and zero the absence of frailty. Frailty as well as pre-frailty are common among persons living with chronic pain which has also been found to be a significant predictor of frailty [27].

**Recent Physical Activity Questionnaire (RPAQ).** To examine the physical activity levels the Recent Physical Activity Questionnaire (RPAQ) was used. This questionnaire assesses physical activity (PA) in 4 domains (leisure, work, commuting, home) during the past month [28]. The home section covers activities, such as computer use, getting about (excluding for work), television, and stair climbing at home. Activity at work covers commuting, type of work, and hours worked per week. Leisure time activities refer to frequently performed activities, such as "mowing the lawn", "exercise with weights", and "jogging". Participants indicated the frequency of the activity and the average time per episode. The English version of RPAQ including the syntax for interpretation is available from: https://www.mrc-epid.cam.ac.uk/physical-activity-downloads/. Summary variables from the RPAQ were derived according to the methods described by Besson and colleagues [29]. Here we used Physical Energy Expenditure (PAEE, [kJ/kg/d]) to denote physical activity.

## Statistical methods

Based on the sample size requirements specified by Fritz and MacKinnon [30], assuming the size of the alpha and beta paths both to be 0.26 (standardised), a minimum sample size of 162 was required for the percentile bootstrap method to be powered at 0.80. Analyses were not split by diagnoses. All analyses were performed using R version 4.4.1 Three analyses were conducted using the R-package "lavaan" [31] to assess the indirect associations between pain resilience and kinesiophobia in the relationship between physical activity and musculoskeletal pain individually and simultaneously. We created 5000 bootstrapped samples to estimate the indirect associations and their standard errors. We tested for statistical significance using a non-parametric boot strapping approach (5000 iterations, $\alpha = 0.05$). We first assessed the direct

and indirect associations of average pain in the last seven days on PAEE through TSK and PRS separately and finally present a model considering TSK and PRS simultaneously (Fig 1).

**Regression analyses.** The lm() function from the "stats" package [32] was used to fit linear models to explore the psychological predictors of physical activity while accounting for musculoskeletal pain cross-sectionally. Prior to conducting the linear regression analysis, correlational analyses (spearman and point biserial) were conducted to examine the bivariate relationships between age, gender, average pain in the last seven days, TSK, PRS, FRAIL, pain duration, and number of pain areas with PAEE. Due to the low sample size, participants reporting non-binary or third gender were excluded from the correlation and regression analyses. Results from the correlational analyses were used to determine whether variables were included in the linear regression model (see Table 1). Diagnostic tests of tolerance and variance inflation revealed all measures fell within acceptable ranges of collinearity (Variance Inflation Factors < 2). Durbin-Watson values were used to check for independence of residuals, which was confirmed. Homoscedasticity was examined by

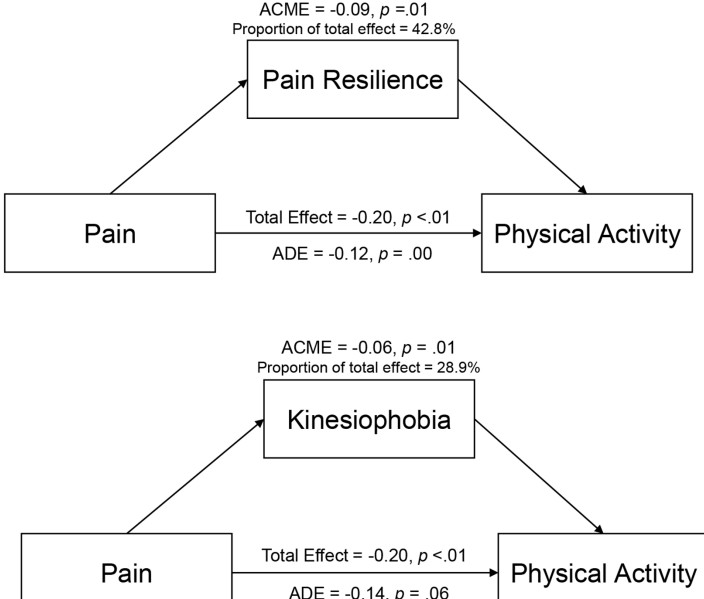

**Fig 1. Overview of direct and indirect associations.**

**Table 1. Correlation Matrix between Demographic, Clinical, and Outcome Measures.**

|  | TSK | PRS | Average Pain last seven Days | Pain duration | No. Pain Areas | Frailty Count | PAEE-tr | Gender |
|---|---|---|---|---|---|---|---|---|
| Age | .03 | .03 | .05 | .24** | .11 | −.03 | −.04 | .03 |
| TSK |  | −.39** | .33** | .10 | .23** | .35** | −.28** | .07 |
| PRS |  |  | −.22** | −.23** | −.19** | −.28** | .40** | .12 |
| Average Pain last seven Days |  |  |  | .20** | .29** | .45** | −.21** | −.07 |
| Pain Duration |  |  |  |  | .20** | .09 | −.21** | −.02 |
| No. Pain Areas |  |  |  |  |  | .32** | −.18* | −.16* |
| Frailty Count |  |  |  |  |  |  | −.24** | −.23** |
| PAEE-tr |  |  |  |  |  |  |  | .27** |

* = *p* < 0.05; ** = *p* < 0.01;

plotting the standardized residuals against the standardized predicted values and the normality of the residuals was checked using histograms. These assumptions were found to be violated. Consequently, the dependent variable (PAEE) was transformed by applying the natural logarithm function (ln), assumptions testing was re-run, and homoscedasticity as well as the normality of the residuals were confirmed. All analyses were run with the transformed version of the PAEE variable, from here on denoted as PAEE-tr. Sample characteristics are provided for the original variable. Following the correlation analyses, gender, average pain in the last seven days, number of pain areas, pain duration, TSK, PRS, and FRAIL were included in the final regression model.

## Results

### Sample characteristics

Table 2 presents the means and standard deviations for participants' physical activity and pain-related psychosocial measures.

### Indirect associations – pain resilience

We tested whether average pain in the last week was indirectly associated with PAEE-tr through pain resilience. The results of the analysis showed that the indirect association (ACME) between average pain in the last week and PAEE-tr through pain resilience was significant (ACME = −0.09, 95% CI [−0.34, −0.06], $p = 0.01$). The average direct association (ADE) between average pain in the last week and PAEE-tr, controlling for pain resilience, was not significant (ADE = −0.12, 95% CI [−0.25, 0.02], $p = .09$). The total association between average pain in the previous week and PAEE-tr ignoring the influence of pain resilience was −0.20 (95% CI [−0.34, −0.06], $p < .01$). Approximately 42.8% of the total association between average pain in the previous week and PAEE-tr indirectly went through pain resilience.

### Indirect associations – kinesiophobia

When testing the influence of kinesiophobia on the relationship between average pain in the previous week and PAEE-tr, the analyses revealed a significant indirect association (ACME, −0.06, 95% CI [−0.12, −0.01], $p = .04$). The ADE, controlling for kinesiophobia, was not significant (−0.14, 95% CI [−0.29, 0.01], $p = .06$). The total association between average pain in the previous week and PAEE-tr not accounting for the influence of kinesiophobia was significant −0.20 (95% CI [−0.34, −0.06], $p < .01$). Approximately 28.9% of the total association between average pain in the previous week and PAEE-tr was accounted for the by the indirect association with kinesiophobia.

### Indirect associations – Pain resilience and kinesiophobia

Finally, we tested the simultaneous association of pain resilience and kinesiophobia with the relationship between average pain in the previous week and PAEE-tr. A significant indirect association was found for pain resilience (ACME, −0.08, 95%

**Table 2. Sample characteristics.**

| Variable | Mean | Median | *SD* | Min | Max |
|---|---|---|---|---|---|
| TSK | 40.4 | 40.0 | *7.2* | 23 | 63 |
| Pain Duration (months) | 78.8 | 48.0 | *89.4* | 3 | 540 |
| Pain Areas | 3.7 | 3.0 | *2.5* | 0 | 13 |
| FRAIL | 1.2 | 1.0 | *1.1* | 0 | 5 |
| Average Pain in the last seven days (0–100) | 44.5 | 41.0 | *19.9* | 5 | 87 |
| PRS | 31.8 | 32.0 | *11.4* | 2 | 56 |
| PAEE | 47.8 | 35.1 | *39.9* | 3.0 | 219.1 |

CI [−0.16, −0.06], $p$ = .01), but not for kinesiophobia (ACME, −0.01, 95% CI [−0.06, 0.04], $p$ = .60). The ADE of average pain in the last week with PAEE-tr, controlling for pain resilience and kinesiophobia, was not significant (ADE = −0.11, 95% CI [−0.24, 0.04], $p$ = .13). The total association between average pain in the previous week and PAEE-tr ignoring the influence of pain resilience and kinesiophobia was −0.20 (95% CI [−0.34, −0.06], $p$ < .01). Approximately 41.1% of the total association of average pain in the previous week and PAEE-tr indirectly went through pain resilience and 6.4% through kinesiophobia.

## Linear regression analysis

We ran a linear regression analysis predicting PAEE-tr. Average pain in the last seven days was entered in the first step. All predictor variables (gender, TSK, PRS, pain duration, number of pain areas, and FRAIL) were entered in the second step of the regression model explaining an additional 22% of the variance in PAEE-tr. Only PRS (β = 0.34, $p$ < .01) and gender (β = 0.23, $p$ < .01) contributed significant unique variance to the prediction of PAEE-tr (see Table 3). This implies that higher PRS scores and male gender were associated with greater PAEE-tr.

## Discussion

Our analyses have demonstrated that significant proportions of the association between musculoskeletal pain and physical activity go through both pain resilience and kinesiophobia. Nevertheless, when considered simultaneously, only the indirect associations via pain resilience remained significant. Further, when predicting physical activity levels, only high levels of pain resilience and male gender were associated with increased physical activity levels – kinesiophobia did not contribute significant unique variance. This highlights the central role pain resilience plays in retaining physical activity levels when faced with chronic pain.

Pain acts as a barrier to physical activity [33,34]. Generally, individuals faced with chronic pain become less physically active, or even sedentary, to avoid pain [3,35]. Nevertheless, pain resilience is the capacity to endure pain and remain active despite pain. Our study may tentatively confirm previous investigations attesting the protective role of resilience in the face of acute and chronic pain [36,37]. Simultaneously, however, the findings may also reflect the negative association between resilience and pain. Nevertheless, pain resilience may be associated with an improved ability to tolerate pain and maintain a positive outlook [25], which may translate into higher physical activity levels. Individuals may employ adaptive coping strategies, such as positive social interactions and adaptive social support-seeking, focus on the long-term benefits of physical activity, and reinterpret pain signals to achieve this, such as by accepting that pain may be outside of their

**Table 3. Regression analysis predicting PAEE-tr.**

| Variables | β | $R^2$ change | F-change | $p$ value |
|---|---|---|---|---|
| Step 1 | | 0.04 | 6.93 (1, 167) | .01 |
| Average pain last seven days | −.20** | | | .01 |
| Step 2 | | 0.22 | 8.16 (6, 161) | <.01 |
| Average pain last seven days | −.06 | | | .43 |
| PRS | .34** | | | <.01 |
| FRAIL | −.02 | | | .80 |
| Number of pain areas | −.03 | | | .73 |
| TSK | −.08 | | | .34 |
| Gender | .23** | | | <.01 |
| Pain duration | −.06 | | | .38 |

Note: N = 169; * $p$ < .05; ** $p$ < .01; Values in parentheses are degrees of freedom

control. [38,39]. They may also be better able to experience reward and positive emotions associated with physical activity despite pain. It is conceivable that pain resilience is associated with reduced avoidance, but the exact mechanisms require further experimental study.

It cannot be ruled out, however, that engaging in physical activity may also have a positive impact on pain resilience. Physical activity can be beneficial in managing certain types of pain, such as arthritis or low back pain [40]. By engaging in physical activity individuals may be able to improve their function and pain or stave off further decline. This experience of pain reduction following physical activity engagement may function as negative reinforcement, making the engagement in physical activity more likely in the future as well as improving pain resilience as a result. Similarly, it cannot be ruled out that pain resilience might just reflect participants having less pain. Therefore, the current findings should be interpreted with caution and warrant further longitudinal and experimental investigation.

Kinesiophobia, on the other hand, increases the likelihood of engaging in avoidance behaviours and thereby reduces engagement in physical activity [12], as was observed in the current study. When pain occurs during physical activity, this acts as a deterrent for future engagement in the form of avoidance behaviours. These are particularly deleterious as the ensuing inactivity may lead to deconditioning, which may further increase pain [33]. The association between physical activity and pain, once established, is difficult to rectify. Due to avoidance of physical activity, it is rarely challenged [41]. In fact, one of the most promising approaches to break this cycle may be to ask individuals to engage in physical activity with the idea that this exposure will lead to corrective experiences (i.e., physical activity in the absence of pain) that will serve individuals to form a new, more adaptive association between pain and physical activity [42]. It may therefore be possible that kinesiophobia may be reduced if physical activity is increased; however, this must be tested experimentally, as the current methodology precludes directional statements.

It is notable that, when considered concurrently, only pain resilience and not kinesiophobia predicted physical activity levels, when accounting for pain, gender, frailty, pain duration, and the number of pain sites. This suggests that pain resilience predicts physical activity levels beyond kinesiophobia and pain itself. This shows also that pain resilience is not merely the opposite to kinesiophobia in this context. In fact, the two concepts were only moderately correlated (r = −0.39). Kinesiophobia is described as the fear of movement-related pain, whereas pain resilience is the ability to cope with and recover from pain [9,25]. As such, kinesiophobia is a fear-based response to pain, associated with negative outcomes. It is also much narrower in its focus on the relationship between pain and movement. Pain resilience on the other hand, is a coping-based response, associated with better health outcomes and reduced pain. Its definition encompasses a broader spectrum of the associations between pain and other factors, such as behaviour, cognition, and emotional responses.

This study also found that men engaged in more physical activity compared to women, echoing previous investigations [43]. This may be rooted in cultural factors and perceived gender norms, as physical activity, in the form of sports, has traditionally been more encouraged and accepted among males than females from an early age [44,45]. Additionally, men tend to have greater self-efficacy in their ability to exercise and experience more social support as well as motivation to engage in physical activity compared to women [46].

It is surprising that frailty did not predict physical activity levels when considered alongside pain resilience and gender. Frailty might partially overlap with pain resilience, such that pain resilience captures aspects of frailty. Indeed, frailer individuals have been found to report lower general resilience [47], but no studies have focused on the association between pain resilience and frailty specifically. Presently, the two concepts were negatively correlated, albeit weakly, suggesting that pain resilience may capture aspects of frailty, but goes beyond this when predicting physical activity levels. Perhaps pain resilience buffers against the negative effects of frailty and so explains why some frail individuals remain active. Longitudinal investigations are needed to establish whether pain resilience can buffer against the development of frailty.

Several limitations warrant caution when interpreting the study's findings. Participants were recruited online, and the sample therefore does not represent a clinical population. Validation of the study's findings in a clinical sample would increase certainty of our conclusions. A number of observations had to be deleted due to incomplete responses, failing

attention checks, or failure to provide accurate data. The instruments employed in this study were validated for use within specific populations, e.g., lower back pain. Given the range of conditions and types of pain present in the current sample, these instruments were, in some cases, used in contexts which they have not yet been validated for. This invites caution when interpreting the results and, crucially, suggest they may not be uniformly applicable to all chronic pain conditions. Finally, the given the cross-sectional nature of the current data, we cannot establish whether pain causes changes in physical activity or vice versa. Experimental studies exploring this issue, considering the role of pain resilience, are needed.

## Conclusion

When considered separately, significant proportions of the associations between musculoskeletal pain and physical activity likely occurred indirectly through both pain resilience and kinesiophobia. Simultaneously, only the indirect path via pain resilience remained significant. Therefore, we can confirm that pain is indirectly associated with physical activity through pain resilience. When accounting for kinesiophobia, frailty, number of pain sites, pain duration, and pain intensity, only pain resilience and gender predicted unique variance in physical activity levels. This implies that pain resilience predicts physical activity levels beyond pain intensity, kinesiophobia, pain duration, and pain spread. It is therefore imperative to examine avenues of increasing pain resilience among individuals suffering from chronic pain, not only to improve their pain, but also their overall health and well-being. Treating pain resilience as a modifiable trait bears implications for clinical practice and may inform treatment approaches, whereby pain resilience may be boosted to increase physical activity levels. Experimental studies that boost pain resilience among persons living with chronic pain are needed to establish the exact nature of the causal relationship between pain resilience, pain, and physical activity.

## Supporting information

**S1 File. STROBE Statement—Checklist of items that should be included in reports of *cross-sectional studies.*** (DOCX)

## Author contributions

**Conceptualization:** Nils Georg Niederstrasser, Nina Attridge.

**Data curation:** Nils Georg Niederstrasser.

**Formal analysis:** Nils Georg Niederstrasser, P. Maxwell Slepian.

**Methodology:** Nils Georg Niederstrasser, Nina Attridge, P. Maxwell Slepian.

**Project administration:** Nils Georg Niederstrasser.

**Visualization:** Nils Georg Niederstrasser.

**Writing – original draft:** Nils Georg Niederstrasser, Nina Attridge, P. Maxwell Slepian.

**Writing – review & editing:** Nils Georg Niederstrasser, Nina Attridge, P. Maxwell Slepian.

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
