## [Decision Letter · Decision Letter 0]

9 Apr 2025

PONE-D-25-08101Indirect Effects of Pain Resilience and Kinesiophobia on the Relationship between Physical Activity and PainPLOS ONE

Dear Dr. Niederstrasser,

Thank you for submitting your manuscript to PLOS ONE. After careful consideration, we feel that it has merit but does not fully meet PLOS ONE’s publication criteria as it currently stands. Therefore, we invite you to submit a revised version of the manuscript that addresses the points raised during the review process.

We look forward to receiving your revised manuscript.

Kind regards,

Cid André Fidelis de Paula Gomes

Academic Editor

PLOS ONE

2. Please include a separate caption for each figure in your manuscript.

Additional Editor Comments:

Thank you for the opportunity to review this manuscript. Please consider the reviewers' comments. Furthermore, I ask that you follow the journal's guidelines and format your manuscript accordingly. In addition, the discussion needs to provide a more detailed description of how the results presented here can help clinical practice.

Reviewers' comments:

Reviewer's Responses to Questions

**Comments to the Author**

1. Is the manuscript technically sound, and do the data support the conclusions?

Reviewer #1: Yes

Reviewer #2: Yes

Reviewer #3: Yes

2. Has the statistical analysis been performed appropriately and rigorously? 

Reviewer #1: Yes

Reviewer #2: Yes

Reviewer #3: No

3. Have the authors made all data underlying the findings in their manuscript fully available?

Reviewer #1: Yes

Reviewer #2: Yes

Reviewer #3: Yes

4. Is the manuscript presented in an intelligible fashion and written in standard English?

Reviewer #1: Yes

Reviewer #2: Yes

Reviewer #3: Yes

5. Review Comments to the Author

Reviewer #1: Dear authors, after reviewing the article, I kindly ask you to make the following adjustments:

-Provide a more detailed explanation of the procedure adopted to determine the sample size

-Please review the caption in table 1. Both “**” should correctly indicate the value of p

-It is recommended that the use of the STROBE guideline be reported and that it be utilized to verify the adherence to the study's methodological steps

Reviewer #2: The article follows the journal's standards, highlighting the statistics used to analyze the results. Where the main points of access to the sample and means of collection are available to the reader.

Reviewer #3: Dear Author,

First of all congratulations for developing the manuscript. Knowledge about pain resilience is a current topic of discussion and a very important theme to be investigated through scientific research. Especially because it goes beyond pain studies.

Title: When you say “effects” you suggest measurements of some intervention (that you must to apply) in some “population”, but you propose by this study an analysis of the associations between the variables. You can also include that you are talking about chronic pain.

Abstract: “This study aimed to examine the direct and indirect impact of pain resilience and kinesiophobia on the relationship between pain and physical activity.” I suggest you use adequate terms such as “examine correlation” insted “examine impact”. “Impact” also sounds like you are developing a clinical trial.

Introduction

It is lacking the objective of the study.

Methods

I suggest you start methods describing what kind of study is this.

Did you classificate the participants' disorders in your analysis?

Were participants recruited without a maximum age limit? Observe that you have a sample characterized with mean age 41.75 but the range is 21-75 years old. Maybe the mean does not represent the sample.

It is not clear the exclusion criteria. You comment that “four participants were excluded due to being outliers for physical activity energy expenditure (PAEE), scoring more than three standard deviations above the mean”, but you must declare the criteria before collecting and analysing data, not after.

It is better to include the information that participants can specify more than one pain location at pain assessment. Was there a limit to this?

Were the participants with and without diagnosis data analysed together? Please describe it.

Did you do a sample size calculation? It must be described.

I assume the data is non parametric because data dispersion is large and also you describe that used the spearman test, so once again why don’t you use median instead mean?

About the tools selected for measurement, observe that each one was validated to a specific population (non specific neck pain, low back pain etc.) and you have applied these indiscriminately because your sample is too generalized. I suggest you include this as a limitation of your study at discussion.

Pain assessment: The reference used (21) does not present the psychometrics properties of numerical rate scale (NRS). It aims at other outcomes. I suggest you explain what kind of tool you used to assess pain (NRS, VAS etc.) and cite a psychometrics properties reference or validation. Once you have this information, it looks better to define a cutoff point, based on your study population, as an inclusion criteria. It is important to be sure that you have people with considerable pain level in your sample.

Tampa Scale for Kinesiophobia: I suggest the same for TSK, if you observe results you have M:40,3 SD7, but there is a minimum 23, do they really have a Kinesiophobia?

Pain Resilience Scale: The sample has M:31,7 SD11,3. How can I interpret this result? Explain if there are cutoff scores and classifications. As well describe how you define if your sample is resilient or not for pain.

FRAIL scale: Here is more completely exposed the tool information. And your sample is classified as pre-frailty, if you consider mean and standard deviation (M:1,2 SD1,1).

Recent Physical Activity Questionnaire (RPAQ): Specify how to interpret scores at this questionnaire. The link mentioned is expired.

Results: The sample characteristics have a very high data dispersion. The range at critical points is too large in all variables. Why don’t you use median instead mean?

How is it possible 0 for pain areas?

It would be visual greater to present some graphics for associations or correlations.

Discussion: You have no discussion in your text. You have wrote a text that can be used as an introduction and repeated results information. A discussion must compare your results with evidence in available literature. Explore more what you have found and what others authors have published, exposing the great points and the limitations of your study and what are the implications for clinicians.

Conclusions

I suggest you be more direct about your conclusion, focused on declaring if your objective was reached. Some sentences used may be moved to discussion. And, for sure you must have adequate terms as in your title and objective.

6. PLOS authors have the option to publish the peer review history of their article (what does this mean?). If published, this will include your full peer review and any attached files.

Reviewer #1: **Yes: **Bruno Ruocco Verengue

Reviewer #2: No

Reviewer #3: **Yes: **Patrícia Gabrielle dos Santos

---

## [Author Response · Author response to Decision Letter 1]

28 May 2025

General comment:

Upon working through the reviewers’ comments, we noticed we had misinterpreted the checks of homoscedasticity and normality of the residuals for the analyses. These assumptions were found to be violated. We therefore applied a natural logarithmic transformation to the dependent variable (PAEE) and were able to establish homoscedasticity as well as the normality of the residuals for all analyses. This did not affect the nature of the results.

Reviewer #1: Dear authors, after reviewing the article, I kindly ask you to make the following adjustments.

Thank you for your suggestions. We have incorporated your suggestions and feel this has greatly improved the manuscript.

1. Provide a more detailed explanation of the procedure adopted to determine the sample size

Thank you for the suggestions. We have added more information pertaining to the sample size determination to the methods section, see pg. 8-9.

2. Please review the caption in table 1. Both “**” should correctly indicate the value of p

Thank you for bringing this to our attention. We have corrected the caption accordingly.

3. It is recommended that the use of the STROBE guideline be reported and that it be utilized to verify the adherence to the study's methodological steps

We have added a completed STROBE document.

Reviewer #2:

We thank reviewer 2 for their comments.

Reviewer #3:

Thank you for your kind words and suggestions. We have amended the manuscript accordingly.

1. Title: When you say “effects” you suggest measurements of some intervention (that you must to apply) in some “population”, but you propose by this study an analysis of the associations between the variables. You can also include that you are talking about chronic pain.

Thank you for the suggestion. We appreciate using the word “effect” in the title may be misleading and have replaced it with “associations”. We also added “chronic” to the title.

2. Abstract: “This study aimed to examine the direct and indirect impact of pain resilience and kinesiophobia on the relationship between pain and physical activity.” I suggest you use adequate terms such as “examine correlation” insted “examine impact”. “Impact” also sounds like you are developing a clinical trial.

We have amended the wording in the abstract to reflect the reviewer’s suggestion.

3. It is lacking the objective of the study.

This has been added. See pg. 4.

4. I suggest you start methods describing what kind of study is this.

We have added the information to the methods section, see pg. 5.

5. Did you classificate the participants' disorders in your analysis?

Participants were asked to report a diagnosis, if known. However, a large proportion of participants reported not knowing their diagnosis, which is common in chronic pain populations. We therefore decided not to use participants’ diagnoses in the analyses.

6. Were participants recruited without a maximum age limit? Observe that you have a sample characterized with mean age 41.75 but the range is 21-75 years old. Maybe the mean does not represent the sample.

We employed a minimum age requirement of 18, but no upper age limit. We aimed to capture a large and representative sample, spanning a wide age range. We have added the median age to the participants section, see pg., 6, to enhance the sample description.

7. It is not clear the exclusion criteria. You comment that “four participants were excluded due to being outliers for physical activity energy expenditure (PAEE), scoring more than three standard deviations above the mean”, but you must declare the criteria before collecting and analysing data, not after.

Identification of outliers as 3SD outside the mean is a common method. Nevertheless, we appreciate removal of outliers may not be warranted in the current study. We therefore re-ran the analyses including the previously omitted cases, observed no significant differences in the results, and therefore decided to include these observations.

8. It is better to include the information that participants can specify more than one pain location at pain assessment. Was there a limit to this?

There was no limit to how many locations participants could specify for their pain. This information has been added, see pg. 6.

9. Were the participants with and without diagnosis data analysed together? Please describe it.

Analyses were not split between participants with and without diagnoses. In the case of chronic pain, diagnoses are often rare and not informative. We have added additional information to the manuscript to explain this, see pg. 10.

10. Did you do a sample size calculation? It must be described.

This has been added, see pg. 8-9.

11. I assume the data is non parametric because data dispersion is large and also you describe that used the spearman test, so once again why don’t you use median instead mean?

Thank you for this suggestion. We have added the median to Table 2 to give additional depth to the representation of our data.

12. About the tools selected for measurement, observe that each one was validated to a specific population (non specific neck pain, low back pain etc.) and you have applied these indiscriminately because your sample is too generalized. I suggest you include this as a limitation of your study at discussion.

This has been added to the limitations section, see pg. 17.

13. Pain assessment: The reference used (21) does not present the psychometrics properties of numerical rate scale (NRS). It aims at other outcomes. I suggest you explain what kind of tool you used to assess pain (NRS, VAS etc.) and cite a psychometrics properties reference or validation. Once you have this information, it looks better to define a cutoff point, based on your study population, as an inclusion criteria. It is important to be sure that you have people with considerable pain level in your sample.

Thank you for pointing this out. Reference 21 was added to demonstrate the validity of using recalled pain ratings to assess pain levels. We have added an additional reference to back up the psychometric properties of using Visual Analogue Scales to assess pain and clarified that a visual analogue scale was used. We believe it is important to capture a range of pain levels and use the data continuously in our analyses, rather than specifying cut-off points.

14. Tampa Scale for Kinesiophobia: I suggest the same for TSK, if you observe results you have M:40,3 SD7, but there is a minimum 23, do they really have a Kinesiophobia?

Pain Resilience Scale: The sample has M:31,7 SD11,3. How can I interpret this result? Explain if there are cutoff scores and classifications. As well describe how you define if your sample is resilient or not for pain.

FRAIL scale: Here is more completely exposed the tool information. And your sample is classified as pre-frailty, if you consider mean and standard deviation (M:1,2 SD1,1).

Recent Physical Activity Questionnaire (RPAQ): Specify how to interpret scores at this questionnaire. The link mentioned is expired.

For the analyses to be valid, it is not necessary for all participants to suffer from kinesiophobia or to be resilient against pain. We are treating these variables as continuous, enabling us to explore the relationships between high and low values with the outcome variables. We are expecting the sample to cover the entire range from low/no resilience to maximum resilience, thus capturing the entire range of response options. This spread allows us to draw more nuanced inferences as to the nature of the relationships between variables.

Apologies for pasting the incorrect link to the RPAQ information. We have updated this.

15. Results: The sample characteristics have a very high data dispersion. The range at critical points is too large in all variables. Why don’t you use median instead mean?

How is it possible 0 for pain areas?

We are using the data as a continuous, capturing the entire range of response options, rather than a categorical measure. Therefore, there are no critical points and the observed variance in scores is not problematic.

Indeed, one participant reported to experience pain in “0” locations. Nevertheless, this was likely because the participant reported suffering from osteoarthritis in the knee and found that none of the locations provided as answer options adequately represented this.

16. It would be visual greater to present some graphics for associations or correlations.

Thank you for this suggestion. After deliberation, we find that the representation of the correlations in the form of a matrix and the direct/indirect effects in the form of the included figures follows that found in similar examples in the literature. We therefore decided not to add additional visual representations to the manuscript.

17. Discussion: You have no discussion in your text. You have wrote a text that can be used as an introduction and repeated results information. A discussion must compare your results with evidence in available literature. Explore more what you have found and what others authors have published, exposing the great points and the limitations of your study and what are the implications for clinicians.

We have made several edits to the discussion section to highlight discussion points, how our findings link with previous research, and added text that further explores our findings.

18. Conclusions

I suggest you be more direct about your conclusion, focused on declaring if your objective was reached. Some sentences used may be moved to discussion. And, for sure you must have adequate terms as in your title and objective.

We have added a sentence to the discussion explicitly stating that the objective was reached. We also changed the terminology to reflect the reviewer’s suggestions.

---

## [Decision Letter · Decision Letter 1]

23 Jun 2025

PONE-D-25-08101R1Indirect Associations of Pain Resilience and Kinesiophobia with the Relationship between Physical Activity and Chronic PainPLOS ONE

Dear Dr. Niederstrasser,

Thank you for submitting your manuscript to PLOS ONE. After careful consideration, we feel that it has merit but does not fully meet PLOS ONE’s publication criteria as it currently stands. Therefore, we invite you to submit a revised version of the manuscript that addresses the points raised during the review process.

We look forward to receiving your revised manuscript.

Kind regards,

Cid André Fidelis de Paula Gomes

Academic Editor

PLOS ONE

Journal Requirements:

**Additional Editor Comments:**

Thank you for the opportunity to review this manuscript. I congratulate the authors on their review.

1. Clarify in the text that the findings represent associations rather than cause-and-effect relationships, and strongly recommend future longitudinal or experimental studies.

2. The sample is extremely heterogeneous, including individuals with different chronic pain conditions, without distinguishing between types of pain. The psychometric tools used were validated for specific populations but were applied indiscriminately to a general chronic pain population. This point should be emphasized in the discussion as a critical limitation, warning that the results may not be uniformly applicable to all chronic pain conditions.

3. A more balanced discussion is advised, acknowledging the fragility of some findings and avoiding excessive generalizations regarding the clinical applicability of the results.

Reviewers' comments:

Reviewer's Responses to Questions

**Comments to the Author**

1. If the authors have adequately addressed your comments raised in a previous round of review and you feel that this manuscript is now acceptable for publication, you may indicate that here to bypass the “Comments to the Author” section, enter your conflict of interest statement in the “Confidential to Editor” section, and submit your "Accept" recommendation.

Reviewer #1: All comments have been addressed

Reviewer #3: All comments have been addressed

2. Is the manuscript technically sound, and do the data support the conclusions?

Reviewer #1: Yes

Reviewer #3: Yes

3. Has the statistical analysis been performed appropriately and rigorously? 

Reviewer #1: Yes

Reviewer #3: Yes

4. Have the authors made all data underlying the findings in their manuscript fully available?

Reviewer #1: Yes

Reviewer #3: Yes

5. Is the manuscript presented in an intelligible fashion and written in standard English?

Reviewer #1: Yes

Reviewer #3: Yes

6. Review Comments to the Author

Reviewer #1: (No Response)

Reviewer #3: Dear author,

The changes adopted in the manuscript were of great relevance for its publication. Congratulations on the work developed.

7. PLOS authors have the option to publish the peer review history of their article (what does this mean?). If published, this will include your full peer review and any attached files.

Reviewer #1: No

Reviewer #3: **Yes: **Patrícia Gabrielle dos Santos

---

## [Author Response · Author response to Decision Letter 2]

3 Jul 2025

Journal Requirements:

We have checked the reference list following the journal’s advice and have not identified any issues.

Editor Comments

1. Clarify in the text that the findings represent associations rather than cause-and-effect relationships, and strongly recommend future longitudinal or experimental studies.

We have made the suggested changes and clarified in the text that the findings are correlational in nature and highlighted the need for further longitudinal and experimental investigation. See for example, pg. 15, ln. 5-6.

2. The sample is extremely heterogeneous, including individuals with different chronic pain conditions, without distinguishing between types of pain. The psychometric tools used were validated for specific populations but were applied indiscriminately to a general chronic pain population. This point should be emphasized in the discussion as a critical limitation, warning that the results may not be uniformly applicable to all chronic pain conditions.

This has been further emphasised in the limitations sections as suggested.

3. A more balanced discussion is advised, acknowledging the fragility of some findings and avoiding excessive generalizations regarding the clinical applicability of the results.

We have adjusted the discussion section in line the suggestions.

---

## [Decision Letter · Decision Letter 2]

30 Aug 2025

PONE-D-25-08101R2Indirect Associations of Pain Resilience and Kinesiophobia with the Relationship between Physical Activity and Chronic PainPLOS ONE

Dear Dr. Niederstrasser,

Thank you for submitting your manuscript to PLOS ONE. After careful consideration, we feel that it has merit but does not fully meet PLOS ONE’s publication criteria as it currently stands. Therefore, we invite you to submit a revised version of the manuscript that addresses the points raised during the review process.

We look forward to receiving your revised manuscript.

Kind regards,

Aynollah Naderi

Academic Editor

PLOS ONE

Journal Requirements:

Reviewer's Responses to Questions

**Comments to the Author**

1. If the authors have adequately addressed your comments raised in a previous round of review and you feel that this manuscript is now acceptable for publication, you may indicate that here to bypass the “Comments to the Author” section, enter your conflict of interest statement in the “Confidential to Editor” section, and submit your "Accept" recommendation.

Reviewer #1: All comments have been addressed

Reviewer #3: All comments have been addressed

Reviewer #4: (No Response)

2. Is the manuscript technically sound, and do the data support the conclusions?

Reviewer #1: Yes

Reviewer #3: Yes

Reviewer #4: Yes

3. Has the statistical analysis been performed appropriately and rigorously? 

Reviewer #1: Yes

Reviewer #3: Yes

Reviewer #4: Yes

4. Have the authors made all data underlying the findings in their manuscript fully available?

Reviewer #1: Yes

Reviewer #3: Yes

Reviewer #4: Yes

5. Is the manuscript presented in an intelligible fashion and written in standard English?

Reviewer #1: Yes

Reviewer #3: Yes

Reviewer #4: Yes

6. Review Comments to the Author

Reviewer #1: Dear authors, congratulations on the work developed in this study. The topic of pain resilience and self-management of chronic conditions remains a complex topic. This study raises hypotheses about the role of cognitive factors in understanding certain musculoskeletal conditions. We look forward to seeing future longitudinal and clinical studies to confirm these associations.

Reviewer #3: (No Response)

Reviewer #4: Please review the comments and suggestions provided in the attached file and incorporate the necessary revisions into the revised version of your manuscript.

7. PLOS authors have the option to publish the peer review history of their article (what does this mean?). If published, this will include your full peer review and any attached files.

Reviewer #1: **Yes: **Verengue, Bruno Ruocco

Reviewer #3: **Yes: **Patricia Gabrielle Dos Santos

Reviewer #4: No

---

## [Author Response · Author response to Decision Letter 3]

22 Sep 2025

Journal Requirements:

We have checked the reference list following the journal’s advice and have not identified any issues.

Reviewer 4

Thank you for the opportunity to review this interesting and timely manuscript. The authors address an important topic by examining the psychological mechanisms underlying the relationship between chronic pain and physical activity, with a focus on pain resilience and kinesiophobia. This is a relevant and clinically meaningful contribution, as understanding protective and maladaptive psychological factors can inform both prevention and intervention strategies.

With minor revisions to clarify methodology, streamline the narrative, and strengthen the discussion of limitations and clinical implications, this paper has the potential to make an important impact in the field.

Thank you for your kind words and encouragement. We have incorporated your suggestions and feel this has greatly improved the manuscript.

Abstract

Some sentences are overly long and complex (e.g., “Significant proportions of the association between musculoskeletal pain on physical activity were found to occur…”). Shorter, more direct sentences would improve clarity.

Consider rephrasing “on the relationship between pain and physical activity” to “in the relationship between pain and physical activity”.

The abstract mentions “three models” but does not specify what type (e.g., mediation analysis, path models). A brief clarification would enhance transparency.

The conclusion is strong but slightly overstated. For balance, add a qualifier about study design (e.g., “Given the cross-sectional design, longitudinal studies are needed to confirm causal pathways”).

Consider emphasizing the potential clinical application of fostering resilience (e.g., through behavioral or psychological interventions).

Thank you for your suggestions. We have adapted the abstract accordingly.

Introduction

The introduction provides a strong conceptual framework and justification for the study, but it would benefit from improved organization, reduced redundancy, and a sharper articulation of the study aim. With minor revisions, it can effectively set up the research question and highlight the study’s contribution.

Minor comments.

Some sentences repeat concepts (e.g., multiple mentions of kinesiophobia leading to activity avoidance). Streamlining these sections would improve readability.

Paragraphs are somewhat long and could be broken into smaller sections to improve clarity, especially the sections on kinesiophobia and pain resilience. For example, break the introduction into three focused paragraphs: (1) physical activity and chronic pain, (2) negative psychological factors (kinesiophobia), and (3) positive psychological factors (pain resilience) leading to the study aim.

The introduction sometimes shifts between broad chronic pain populations and specific conditions (e.g., osteoarthritis, fibromyalgia, low back pain). Consider clearly specifying whether the study targets a particular population or general chronic pain sufferers.

The transition from negative psychological factors (kinesiophobia) to positive factors (pain resilience) is slightly abrupt. A bridging sentence explaining why examining positive factors complements existing research would improve flow. Smoothly transition from negative to positive psychological factors, emphasizing how examining both provides a fuller understanding of the pain–activity relationship.

A few sentences are slightly wordy or complex. For example, “It is, however, not clear if this extends to affecting the association between pain and physical activity” could be simplified to “It remains unclear whether this also influences the pain–physical activity relationship.”

Consider highlighting the novelty or practical significance of investigating pain resilience as a modifiable factor.

Thank you for your suggestions to improve our introduction. We have actioned all your suggestions with one exception, please see below.

The final sentence introduces the study aim but could be more concise and explicit. For example, clearly stating: “This study aims to investigate whether pain resilience mediates the relationship between chronic pain and physical activity” may strengthen clarity.

We specifically decided not to label this as mediation, as the term mediation implies a causal relationship that we can not infer given the limitations of the cross-sectional study design.

Methods

The Methods section is thorough and provides sufficient detail for replication. However, improvements in organization, conciseness, and clarity would enhance readability and focus. With minor revisions, this section would clearly communicate the study design, participants, measures, and statistical approach to the reader.

Minor inconsistencies in percentages (e.g., 86 females (50.3%) vs. 83 males (48.3%) with 2 non-binary/third gender (1.2%) and 1 not reported (0.6%) adds to >100%) should be double-checked for accuracy. Ensure all percentages and participant characteristics add correctly to 100%.

Thank you for pointing out this oversight. 86 out of 172 is indeed 50.0%. We have amended this in the manuscript. Please note, the numbers still do not add up to exactly 100%, but this is due to rounding to one decimal.

While all measures are described, the rationale for including some (e.g., FRAIL scale) in this context could be made clearer. How does frailty relate to chronic pain, resilience, or physical activity in this study? Provide a brief justification for including the FRAIL scale and its relevance to the study outcomes.

Frailty reflects vulnerability to adverse health outcomes, which are highly relevant in the context of chronic pain 1. Frailty has been linked to higher pain prevalence, lower resilience, and reduced capacity for physical activity, making it a meaningful construct for understanding variability in pain experiences and rehabilitation outcomes in this population. We have added a brief justification to the manuscript and a reference to 1.

The section could briefly mention whether assumptions for the statistical models (e.g., normality, linearity) were checked.

Thank you for suggesting this. We have reported assumptions testing in the statistical methods section, e.g., assessing the normality of the residuals.

Discussion

The text repeatedly notes that mechanisms are “unclear” or “cannot be ruled out.” Stronger, more specific hypotheses or a conceptual model would add value.

Thank you for your suggestion. The cautious language was requested by previous reviewers and editors to highlight the fact that causality cannot be established in the current study.

The gender-related findings are mentioned but cultural explanations are very brief. More evidence or references on cultural and social aspects would strengthen the argument.

The overlap between frailty and pain resilience is only briefly mentioned. A deeper discussion or concrete suggestion for future research would improve this section.

Several grammatical errors and awkward phrases (e.g., “kinesiophobia was not contribute significant unique variance”) reduce readability. Careful English editing is needed.

The section ends with limitations but lacks a strong concluding paragraph summarizing the central findings, clinical relevance, and need for longitudinal studies.

We have actioned all your suggestions. We have added an additional reference 2.

References:

1. Lin, T., Zhao, Y., Xia, X., Ge, N. & Yue, J. Association between frailty and chronic pain among older adults: a systematic review and meta-analysis. Eur Geriatr Med 11, 945–959 (2020).

2. Mathew Joseph, N., Ramaswamy, P. & Wang, J. Cultural factors associated with physical activity among U.S. adults: An integrative review. Applied Nursing Research 42, 98–110 (2018).

---

## [Editor Report · Decision Letter 3]

24 Sep 2025

Indirect Associations of Pain Resilience and Kinesiophobia with the Relationship between Physical Activity and Chronic Pain

PONE-D-25-08101R3

Dear Dr. Niederstrasser,

We’re pleased to inform you that your manuscript has been judged scientifically suitable for publication and will be formally accepted for publication once it meets all outstanding technical requirements.

Kind regards,

Aynollah Naderi

Academic Editor

PLOS ONE
---

## [Editor Report · Acceptance letter]

PONE-D-25-08101R3

PLOS ONE

Dear Dr. Niederstrasser,

I'm pleased to inform you that your manuscript has been deemed suitable for publication in PLOS ONE. Congratulations! Your manuscript is now being handed over to our production team.

Kind regards,

on behalf of

Dr. Aynollah Naderi

Academic Editor

PLOS ONE